# Size matters: An analysis of cigarette pack sizes across 23 European Union countries using Euromonitor data, 2006 to 2017

May C. I. van Schalkwyk[1,2]*, Martin McKee[2], Jasper V. Been[3,4,5], Christopher Millett[1], Filippos T. Filippidis[1]

**1** Public Health Policy Evaluation Unit, School of Public Health, Imperial College London, London, United Kingdom, **2** Faculty of Public Health and Policy, London School of Hygiene and Tropical Medicine, London, United Kingdom, **3** Department of Paediatrics, Division of Neonatology, Erasmus MC – Sophia Children's Hospital, University Medical Centre Rotterdam, Rotterdam, Netherlands, **4** Department of Public Health, Erasmus MC – Sophia Children's Hospital, University Medical Centre Rotterdam, Rotterdam, Netherlands, **5** Usher Institute, The University of Edinburgh, Edinburgh, United Kingdom

* may.vanschalkwyk@lshtm.ac.uk

## Abstract

### Introduction

The tobacco industry (TI) has used small cigarette pack sizes to encourage brand-switching and consumption, and to mitigate the impacts of tobacco tax increases. Since 2016, the European Union (EU) Tobacco Products Directive (TPD) specifies a minimum pack size of 20 cigarettes. We examined cigarette pack sizes in the EU and whether pack size composition differed between cheap and expensive price segments, as well as the impact of the revised TPD.

### Methods

We conducted a longitudinal analysis of pricing data from 23 EU countries between 2006–2017. We examined pack sizes over time to assess the impact of the TPD, differences in pack size composition between cheap and expensive price segments, and compared gaps in median prices between products using actual and 'expected' prices (price if all packs contained 20 sticks).

### Results

Cigarette pack sizes changed over time, across the EU. The distribution of pack sizes varied between price segments, with small pack sizes especially frequent in the cheap segment of the cigarette market, but this varied over time and across countries. Packs of <20 cigarettes almost disappeared from the data samples after implementation of the TPD.

### Conclusion

Implementation of the TPD appears to have virtually eliminated packs with <20 cigarettes, restricting their use by the TI. Our analysis suggests pack sizes have been used

**Data Availability Statement:** The data underlying the results presented in the study are commercially available from Euromonitor International

https://www.portal.euromonitor.com/portal/magazine/homemain.

**Funding:** JB is supported by personal fellowships from the Netherlands Lung Foundation and Erasmus MC. The Public Health Policy Evaluation Unit is grateful for support from the NIHR School of Public Health Research. Role of the Funder: The funding sources had no role in the design and conduct of the study; collection, management, analysis, and interpretation of the data; preparation, review, or approval of the manuscript, or the decision to submit the manuscript for publication.

**Competing interests:** The authors have declared that no competing interests exist.

differentially across the EU. Country-level analyses on the industry's use of pack sizes, consumer responses, and evaluations of restricting certain pack sizes are needed to confirm our findings and strengthen policy.

## Introduction

Effective action against smoking-related disease requires a detailed understanding of strategies employed by the tobacco industry (TI) to sustain smoking rates, [1–3] including varying pack sizes, both small (e.g. 10 cigarettes) and large (e.g. 35), to segment the market. [3] Thus, Marlboro 10s, an iconic but expensive brand when sold in larger packs, are perceived as "convenient, unique and affordable by young adults". [4] Yet there is little independent research on consumer responses to pack size. [3,5]

High purchase price of a cigarette pack reduces tobacco consumption, especially among those with low disposable income, explaining why sticks are still sold individually in many low-income countries. [6] In the UK, between 2009 and 2015, the TI varied pack sizes, [1] with more cheap forms of manufactured and roll-your-own (RYO) cigarettes and introduction of smaller pack sizes (17–19 stick and 10g packs, respectively). [1] This ensured that cheap products would remain available despite considerable tax increases, [1] undermining the main objective of this highly effective tobacco control policy. [7,8]

More smokers, particularly the poorest, now consume cheaper forms of manufactured or RYO cigarettes. This complicates analyses based on standardised 20-stick pack prices, which may not reflect real-world price differentials. Conversely, for those able to afford a higher upfront cost, cartons of cigarettes enable a price-per-stick saving, again undermining the deterrent effect of price increases. [9] One Australian study concluded that "[V]ariation in pack size remains a powerful form of promotion". [10]

The European Union (EU) Tobacco Products Directive (TPD) specifies a minimum pack size of 20 cigarettes, [11] having come into force on the 19th May 2014 and subsequently applicable in all EU countries since the 20th May 2016. However, other than a few studies in individual countries, [12] little is known about how pack sizes changed across the EU prior to harmonisation. This information is important, as the TI can be expected to seek other ways of circumventing TPD goals. These data can also inform evaluation of this policy change and provide a baseline for future research. Additionally, the European experience can inform policies elsewhere. We therefore describe (1) differences in pack size composition between cheap and expensive price segment across the EU and (2) compliance with minimum pack size following implementation of the TPD.

## Data and methods

We purchased commercial data from Euromonitor International, a private market research company reporting annual product and country-specific information, including on tobacco products sold in EU countries. Analysts record details of multiple tobacco products, aiming to cover at least the 10 brands with the highest market share per country and year. The details of multiple products within a brand are recorded and provided by Euromonitor. For each product, pack size, packs per item, brand and company, and price in local currency, are recorded. The initial dataset contained 34,207 product observations from 23 EU countries spanning 2006–2017. Euromonitor does not routinely collect tobacco data from Croatia, Cyprus, Malta, and Luxembourg. Austria was not included in the analysis as data were only available from

2016 onwards. Data for 2007 from Belgium and 2006 from Slovakia were missing. Ethical approval was not required.

We conducted a two-staged analysis: a 'pooled' analysis using data from all countries to explore trends at the EU level across time, followed by an analysis with data separated by country and year, to explore trends at the country level. All analyses were assessed graphically.

Each product was assigned to one of three pack size categories, <20, 20 or >20 sticks, based on recorded pack size. Mean percentage of each pack size category per year was calculated, allowing each country to be weighted equally. Pack size composition as a percentage of the annual sample was then presented graphically.

The distributions of actual (real-world prices faced by the consumer) cigarette pack prices for each year were then arranged in ascending order and divided into quartiles. The products within the lowest and upper-most quartiles were designated as '*cheap*' and '*expensive*' products respectively. The percentage share of each price quartile, made up by each pack size category, was presented graphically. We then repeated the same analysis for each country included in the study.

Analyses were based on annually-recorded pack size and price data only. This approach allowed us to use the data available to analyse differences in pack sizes in cheap and expensive quartiles in each country over the study period.

We also explored the impact of using '*expected*' (i.e. had all products contained 20-sticks) versus *actual* pack prices on the calculated difference in median price of cheap and expensive cigarettes. [1,2,13] First, we calculated the difference in median price of cheap and expensive quartiles based on the actual price described above. Then, we calculated the expected price (by dividing the actual price by the pack size and multiplying by 20 thereby standardising all prices to those that would be expected had all products contained 20 sticks). We then followed the same approach using this calculated expected price. That is, products in the lowest quartile were designated as '*cheap*' and those in the upper most quartile as '*expensive*' and measured the difference in median prices between these two quartiles. If the difference in median price is larger when using actual rather than expected pack prices, then comparisons based on expected 20-stick packs (a conventional method used in research analyses of tobacco taxation and prices) [13] will *underestimate* the size of the real-world price gap faced by consumers.

All prices were adjusted for inflation to a baseline year of 2015 and converted to Euros as described previously. [14]

## Results

While 20-packs were the norm in most years (the mean percent ranged from 67% to 91% during the study period), the variety of cigarette pack sizes in annual samples changed over the study period at the EU level (Fig 1). Pack size composition in any given year differed between the cheap and expensive quartiles, with <20-packs comprising a greater percentage of the cheap quartile. The mean percentage of <20-packs peaked at 37% in 2012 within the cheap quartile but were then virtually eliminated (1%) across the EU market by 2017 and after the introduction of the TPD. Minimal changes were observed in the percentage of the annual samples composed of large pack sizes.

Similar results were observed at the county level, with 20-packs being the norm in most years and countries. The pack size composition in annual samples changed over the study period, both within and between countries (S1 Fig). Furthermore, differences in pack size composition between price quartiles (based on actual price) were observed, which varied among years and countries. Nineteen of 23 countries had pack sizes of <20 cigarettes in the market during the study period. In 9 of them, the <20 pack size category was exclusively in the

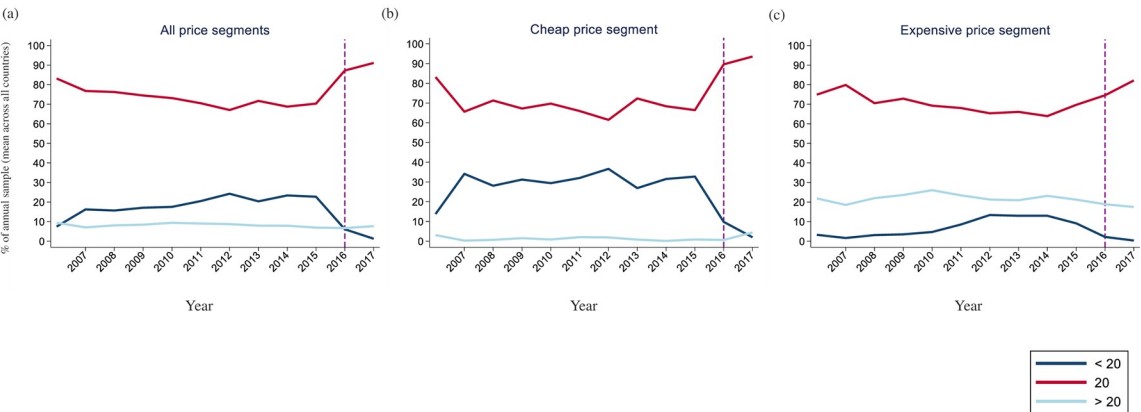

**Fig 1. Pack size category as a percentage of the annual sample over time (a) pack size composition across all price quartiles (b) pack size composition of the cheap quartile (c) pack size composition of the expensive quartile.** The dashed reference bar indicates the year of introduction of the TPD.

cheap segment. Conversely, pack sizes >20 were more likely to be in the expensive segment, where they made up a larger proportion of products. Under-20 pack sizes were not observed in four countries (France, Estonia, Lithuania, and Poland). 20-packs were sold in all countries and in all price segments at some point between 2006 and 2017. In some countries, for example, Germany, the Netherlands and the UK, the entire cheap segment comprised <20 pack sizes in multiple years, whereas in the Czech Republic <20 packs comprised at most 3% in the cheap segment. Differences were similarly observed for the dominance of large pack sizes within any given market and year. For example, larger pack sizes (>20) were the dominant pack-size category observed in the expensive quartile in most years in some of the countries.

By 2017 <20 packs had disappeared from all but 3 countries (Belgium, Bulgaria, Romania), although even there they comprised a small proportion.

When the composition of pack size categories differed between cheap and expensive quartiles, the estimated median prices and, in turn the estimated price gaps between quartiles, differed depending on whether actual or expected prices were used. That is, when the cheap quartile contains products with <20 sticks and the expensive quartile, products with >20 sticks, or vice versa (meaning the pack size composition differs between price segments in the market of analysis), the use of expected 20-stick pack prices (i.e. conversion of all product prices to a standard 20-stick pack price) alters the estimated magnitude of the price gap (data not shown).

## Discussion and conclusions

Consistent with prior research in the UK, [1,2] we observed that cigarette pack sizes sold in 23 EU countries varied between markets, price quartiles, and over time. While <20, 20 and >20 stick packs were observed in both the expensive and cheap quartiles, smaller pack sizes (<20 sticks) featured predominantly in the cheap quartile, likely representing TI tactics to ensure that upfront cost of packs remained affordable and potentially undermining the intended effects of tobacco tax increases. Implementation of the TPD appears to have dramatically reduced the presence of packs with <20 cigarettes across the EU market.

Internal industry documents show that the TI understands the nuances of using different package quantities to undermine tax increases and target certain consumer groups. [3] The introduction of legislation mandating minimum pack sizes, such as the EU TPD, can limit the

use of such strategies, and potentially reduce consumption and initiation. Indeed, we found that packs with fewer than 20 sticks were virtually eliminated from the EU market after 2016. Our findings may therefore help to inform policies in the many countries lacking such legislation. Our research might also contribute to the understanding of the role of smaller pack sizes. Although small packs may help some smokers with higher intention to quit to ration consumption, [3,5] availability of 'cheap' products may hinder smoking cessation or encourage experimentation with tobacco among youth and less affluent individuals. Importantly, however, the impact of banning or mandating certain pack sizes remains poorly researched, and further research is needed to confirm our findings.

Differences in upfront prices between packs of expensive and cheap cigarettes are likely to be underestimated when prices are converted to expected 20-stick pack prices or only 20-stick products are sampled for research and monitoring purposes. [13,15,16] Similarly, Liber *et al* note that the use of expected 20-stick pack prices masked the 'volume discount' afforded to smokers of international versus domestic cigarettes in Southeast Asia. [17] Although the use of expected 20-stick pack prices supports comparative analyses across products and time, this should be complemented by gauging the extent to which this leads to under- or over-estimation of price differences between segments. By converting actual prices to those that would be expected if all packs contain 20 sticks, researchers are essentially 'standardising-out' the effect that the differences in pack size composition between price segments has on the real-world price gaps that consumers face. Accurate assessment is important as large price differences undermine tobacco control efforts by negatively influencing quit attempts and smoking initiation, and have been associated with poor health outcomes. [13] It also allows for the identification of tax shifting, including over-shifting (increasing product prices above and beyond the tax). Over-shifting can lead to increases in price gaps as well as industry profits which in turn may be seen as a threat to tobacco control efforts as such resources can serve to maintain the industry's power to fight public health tobacco control measures. [18]

While Euromonitor data is intended for commercial use, its pricing and packaging data have been increasingly used in public health research. [13,19] Variation in sampling methods would be unlikely to fully explain such clear differences in pack sizes between segments, and our findings in the UK agree with previous analyses of that market. However, we cannot be certain if samples are fully representative of national markets. It is also beyond the scope of this study to assess the impact of pack size and affordability on consumer behaviour. The use of commercial pricing rather than sales data, meant we were unable to explore price gaps using the weighted average price (a measure based on sales volume) or industry-defined price segments, as in previous studies. [1] Small sample sizes and a single time point when data were collected in each year and country precluded us from drawing conclusions about individual markets and years, and from quantifying changes in pack and stick price within markets and price segments. We can therefore only make high-level conclusions about general observations seen across the EU and over time. It is therefore imperative that the current study be viewed as providing a preliminary analysis across the EU region as a whole, exposing strategies potentially adopted by the TI in need of further investigation. Comprehensive country-specific studies using sales and price-promotions data are needed to further evaluate and understand: (1) pack size use by the TI as a pricing and targeting strategy, (2) its relation to consumer behaviour, and (3) the impact of regulating pack sizes. This information is needed to critique information on the impact of pack sizes disseminated by TI-funded bodies such as Change Incorporated (funded by Philip Morris International). [20]

In conclusion, these results show the scale and nature of varying pack sizes across the EU. Our scoping study suggests that pack sizes of <20 appear to have been virtually eliminated from EU markets, strongly suggesting that the TPD policy was effective. Pack sizes varied

differentially between cheap and expensive price quartiles and between countries and years. This study provides further impetus for a greater focus on tobacco product pack size. Specifically, there is a need for a thorough evaluation of industry strategies, consumer responses to pack sizes, and the impact of regulating the use of small pack sizes on smoking rates and consumption both within and beyond Europe, particularly in those countries where the sale of single cigarettes and small pack sizes remains common.

## Supporting information

**S1 Fig. Pack size category as a percentage of cheap (left-hand graphs) and expensive (right-hand graphs) quartiles based on actual pack prices per country-year.** Pricing data was not available for every year in all countries (see methods for details). An expensive quartile of the market was not available for 2008 Denmark and 2016 Romania as four distinct price quartiles could not be calculated for these annual price distributions.
(DOCX)

## Author Contributions

**Conceptualization:** May C. I. van Schalkwyk, Martin McKee, Filippos T. Filippidis.

**Data curation:** May C. I. van Schalkwyk, Martin McKee, Jasper V. Been, Christopher Millett, Filippos T. Filippidis.

**Formal analysis:** May C. I. van Schalkwyk, Martin McKee, Filippos T. Filippidis.

**Methodology:** May C. I. van Schalkwyk, Martin McKee, Filippos T. Filippidis.

**Writing – original draft:** May C. I. van Schalkwyk, Martin McKee, Jasper V. Been, Christopher Millett, Filippos T. Filippidis.

**Writing – review & editing:** May C. I. van Schalkwyk, Martin McKee, Jasper V. Been, Christopher Millett, Filippos T. Filippidis.

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
