## [Decision Letter · Decision Letter 0]

27 Jan 2020

PONE-D-19-33505

Size matters: an analysis of cigarette pack sizes across 23 European Union countries using Euromonitor data, 2006 to 2017

PLOS ONE

Dear Dr van Schalkwyk,

Thank you for submitting your manuscript to PLOS ONE. After careful consideration, we feel that it has merit but does not fully meet PLOS ONE’s publication criteria as it currently stands. Therefore, we invite you to submit a revised version of the manuscript that addresses the points raised during the review process.

Please be sure to address Reviewer 1's concerns about the need to comply with PLOS ONE's data access policy.

We would appreciate receiving your revised manuscript by Mar 12 2020 11:59PM. To enhance the reproducibility of your results, we recommend that if applicable you deposit your laboratory protocols in protocols.io, where a protocol can be assigned its own identifier (DOI) such that it can be cited independently in the future. For instructions see: http://journals.plos.org/plosone/s/submission-guidelines#loc-laboratory-protocols

We look forward to receiving your revised manuscript.

Kind regards,

Stanton A. Glantz

Academic Editor

PLOS ONE

Journal Requirements:

Reviewers' comments:

Reviewer's Responses to Questions

**Comments to the Author**

1. Is the manuscript technically sound, and do the data support the conclusions?

Reviewer #1: Partly

Reviewer #2: Yes

2. Has the statistical analysis been performed appropriately and rigorously? 

Reviewer #1: Yes

Reviewer #2: Yes

3. Have the authors made all data underlying the findings in their manuscript fully available?

Reviewer #1: No

Reviewer #2: Yes

4. Is the manuscript presented in an intelligible fashion and written in standard English?

Reviewer #1: Yes

Reviewer #2: No

5. Review Comments to the Author

Reviewer #1: The authors present a concise study of cigarette pack size and pricing across EU countries using commercial market research data. Use of commercial data such as this gives valuable insights into both tobacco industry strategies, and by proxy, consumer purchasing preferences. The authors provide a balanced description of the strengths and limitations of this data, and how this research fits into the broader picture of research needed to understand tobacco product and pricing strategies. The aims of the study are important and my suggestions are designed to encourage the authors to make the findings of this rich data source more accessible and generalisable.

The authors have analysed a huge amount of data: 23 countries, 11 years per country (mostly), and at least 10 brands per years and country. Two important aims are identified: to describe “differential use of cigarette pack sizes by price segment across the EU” and “compliance with minimum pack size following implementation of the TPD”. The first aim mentions differential use of pack sizes. Use is an ambiguous term here: does it refer to a tobacco industry strategy, or consumer use – in which case, it would be important describe the market share coverage of the top 10 brands if possible. It is also unclear whether the Euromonitor data identifies the top 10 brands, then includes the details of all products within that brand (e.g. multiple variants and pack size combinations). Or is it 10 individual products (stock-keeping units) per country-year?

The current approach to presenting the results does not comprehensively address the aims of the study – clearly an artefact of the longitudinal data from many countries. The current graphs are quite cluttered and small, and the stacked histogram format makes it difficult to compare proportions across years and countries. It might not be necessary to present data for all years – perhaps every second or third year would suffice. It would also more clearly demonstrate the findings of the second aim if you were to annotate on the graph when the TPD was implemented. Including only the lower and upper quartiles of prices is also limiting. At present, the reader cannot examine the pack size composition of the whole market in any country or year – for how many years and countries were no packs of <20 observed prior to the TPD?. Alternatively, country-level data could be presented as supplementary tables and counties could be grouped into meaningful categories, perhaps by price or proportion of small packs.

The authors could consider whether it would be more informative to present the % of products that were in each pack size category, then the % of the cheapest and most expensive quartiles that were made up of packs of <20 and >20 cigarettes. This would also very clearly demonstrate the data necessary to assess the second aim of the study. It would also be useful to more clearly see whether there was a change in the proportion of large packs after the TPD. Did the TI maintain dispersion by introducing more large packs to the market when the lower-priced small packs were eliminated?

The supplementary graph is similarly difficult to review, and perhaps the figure title could be modified to guide the reader through this data. A more detailed explanation of these findings is needed in the results. Overall, the distribution of difference between the actual and expected prices looks quite similar, with a few notable exceptions. The graph does not allow for easy comparison of the magnitude of differences. It is also difficult to assess whether these large differences correspond to countries and years that had higher proportions of non-20 pack sizes from Figure 1.

Some additional points of clarification would be useful:

For a non-EU audience, please briefly describe how the pre- and post-2004 countries differ on factors that might relate to tobacco pricing.

Was the TPD pack size directive implemented on 1 January 2016, or was it phased in?

Page 9: Please elaborate how price differences act to undermine tobacco control efforts, beyond the poor health outcomes associated with tobacco use.

Reviewer #2: This paper confirms that standardizing pack sizes in the EU was an effective strategy to block promotion through smaller pack sizes. However the authors have tried to pack too much information into the accompanying graphs, making them unreadable.

The authors may wish to make them readable by reducing the amount of information displayed. It is suggested that date for just three years at the beginning middle and end of the study period be displayed. Rather than displaying data for all countries, the point could be made by showing data for one country from each sub-region

6. PLOS authors have the option to publish the peer review history of their article (what does this mean?). If published, this will include your full peer review and any attached files.

Reviewer #1: No

Reviewer #2: No

---

## [Author Response · Author response to Decision Letter 0]

28 Jun 2020

Reviewer #1: The authors present a concise study of cigarette pack size and pricing across EU countries using commercial market research data. Use of commercial data such as this gives valuable insights into both tobacco industry strategies, and by proxy, consumer purchasing preferences. The authors provide a balanced description of the strengths and limitations of this data, and how this research fits into the broader picture of research needed to understand tobacco product and pricing strategies. The aims of the study are important and my suggestions are designed to encourage the authors to make the findings of this rich data source more accessible and generalisable.

We would like to thank the reviewer for their thoughtful feedback and have attempted to address each of the specific points made.

The authors have analysed a huge amount of data: 23 countries, 11 years per country (mostly), and at least 10 brands per years and country. Two important aims are identified: to describe “differential use of cigarette pack sizes by price segment across the EU” and “compliance with minimum pack size following implementation of the TPD”. The first aim mentions differential use of pack sizes. Use is an ambiguous term here: does it refer to a tobacco industry strategy, or consumer use – in which case, it would be important describe the market share coverage of the top 10 brands if possible. It is also unclear whether the Euromonitor data identifies the top 10 brands, then includes the details of all products within that brand (e.g. multiple variants and pack size combinations). Or is it 10 individual products (stock-keeping units) per country-year?

We have removed the term ‘use’ and now refer to differences in pack size composition within price segments throughout the text, most notably where the term is first mentioned in the Abstract and the main text:

Abstract:

Introduction:

We examined cigarette pack sizes in the EU and whether pack size composition differed between cheap and expensive price segments, as well as the impact of the revised TPD.

(Page 2 lines 25-32)

Methods:

We conducted a longitudinal analysis of pricing data from 23 EU countries between 2006-2017. We examined pack sizes over time to assess the impact of the TPD, differences in pack size composition between cheap and expensive price segments, and compared gaps in median prices between products using actual and ‘expected’ prices (price if all packs contained 20 sticks).

(Page 2 lines 34-39)

Introduction:

We therefore describe (1) differences in pack size composition between cheap and expensive price segment across the EU and (2) compliance with minimum pack size following implementation of the TPD.

(Page 5, lines 100-103)

We have added the following sentence to the Methods section, elaborating on the product details that are recorded by Euromonitor:

The details of multiple products within a brand are recorded and provided by Euromonitor.

(Page 5, lines 109-110)

The current approach to presenting the results does not comprehensively address the aims of the study – clearly an artefact of the longitudinal data from many countries. The current graphs are quite cluttered and small, and the stacked histogram format makes it difficult to compare proportions across years and countries. It might not be necessary to present data for all years – perhaps every second or third year would suffice. It would also more clearly demonstrate the findings of the second aim if you were to annotate on the graph when the TPD was implemented. Including only the lower and upper quartiles of prices is also limiting. At present, the reader cannot examine the pack size composition of the whole market in any country or year – for how many years and countries were no packs of <20 observed prior to the TPD?. Alternatively, country-level data could be presented as supplementary tables and counties could be grouped into meaningful categories, perhaps by price or proportion of small packs.

The authors could consider whether it would be more informative to present the % of products that were in each pack size category, then the % of the cheapest and most expensive quartiles that were made up of packs of <20 and >20 cigarettes. This would also very clearly demonstrate the data necessary to assess the second aim of the study. It would also be useful to more clearly see whether there was a change in the proportion of large packs after the TPD. Did the TI maintain dispersion by introducing more large packs to the market when the lower-priced small packs were eliminated?.

We thank the reviewer for these very helpful suggestions. We considered both reviewers’ comments simultaneously when revising the analyses and figure, and how best to convey the main messages of the paper. As reviewer one discusses, to answer if <20 packs have been eliminated from the market it is important to look across all price segments. We have therefore undertaken additional grouped analyses at the EU level including (1) all price quartiles, and (2) separately for the cheap and expensive price quartiles to meet both aims of the study – to research the impacts of the TPD and explore for differential pack size composition between price segments. 

Specifically, we conducted a two-staged analysis: a new grouped analysis (as suggested by the first reviewer) using data from all countries to explore trends at the EU level across time, followed by our original analysis with data separated by country and year, to explore trends at the country level. The results are presented graphically displaying the mean percentage of each pack size category in line with the first reviewers’ recommendations. We have also drawn attention to the trends in the large pack size category (>20 cigarettes) in the Results section (Page 7, lines 171-172).

The results of the EU level analysis are presented as Figure 1 (replacing the original Figure 1 of country-level analyses) and the original country-level analyses are now included as Supplementary Material. We have removed the original figures from the Supplementary Material (differences in price gaps when comparing cheap and expensive price segments). Based on the feedback from both the reviewers, we believe that the original Supplementary Material did not add to the overall messages of the paper. As advised by the first reviewer, Figure 1 has a reference bar in each of the graphs indicating the year the TPD was introduced.

We have kept both analyses, the new grouped analysis as well as the original country-level analysis, as we believe the results of both these analyses complement each other and provide a more comprehensive picture of the high-level trends for the reader. The country-level figures have been further edited (i.e. fewer year labels are included) to make the picture more interpretable by, and visually appealing to, the reader.

The text in the Methods and Results now reflects the inclusion of these additional analyses and changes to the content of Figure 1 and Supplementary Material.

The supplementary graph is similarly difficult to review, and perhaps the figure title could be modified to guide the reader through this data. A more detailed explanation of these findings is needed in the results. Overall, the distribution of difference between the actual and expected prices looks quite similar, with a few notable exceptions. The graph does not allow for easy comparison of the magnitude of differences. It is also difficult to assess whether these large differences correspond to countries and years that had higher proportions of non-20 pack sizes from Figure 1

We have removed the original content of the Supplementary Material. Based on both the reviewers’ comments, we feel that the figure does not strengthen the paper and is not useful to the reader. Our aim is to raise the point that differences in pack size composition between price segments, when present in the market, will affect the price gap between cheap and expensive products. If product prices are then converted into standard 20-stick pack prices, which is often done in the academic literature, the median price gap that is measured will be over or underestimated. This section of the results now reads (new text in bold):

That is, when the cheap quartile contains products with <20 sticks and the expensive quartile, products with >20 sticks, or vice versa (meaning the pack size composition differs between price segments in the market of analysis), the use of expected 20-stick pack prices (i.e. conversion of all product prices to a standard 20-stick pack price) alters the estimated magnitude of the price gap (data not shown).

(Page 8, lines 211-216)

The following sentence was deleted from the Methods section:

Finally, we present graphically the difference in median prices for actual and expected pack sizes.

(Page 6, Line 148)

Some additional points of clarification would be useful:

For a non-EU audience, please briefly describe how the pre- and post-2004 countries differ on factors that might relate to tobacco pricing.

We thank the reviewer for this comment. Although there are some contextual differences between pre- and post-2004 countries, we have decided that splitting EU member states in these two groups adds to the complexity of the text without offering much additional information to the reader. Thus, in an effort to simplify our text to clearly convey the main messages we have deleted the following sentence from the Data and Methods section:

We separated countries based on year of EU accession (pre- and post-2004), as in previous studies, given differences in economic development and regulatory environments.

(Page 5, lines 115)

We have also edited the Results section to delete references to pre- and post-2004 countries. The relevant text now reads:

Under-20 pack sizes were not observed in four countries (France, Estonia, Lithuania, and Poland).

(Page 7, lines 181-182)

For example, larger pack sizes (>20) were the dominant pack-size category observed in the expensive quartile in most years in some of the countries.

(Page 8, lines 198-199)

The figure containing the country-level analyses (new Supplementary Material) is no longer separated into pre and post-2004 countries.

 Was the TPD pack size directive implemented on 1 January 2016, or was it phased in?

We have added the following to the Introduction (new text in bold):

The European Union (EU) Tobacco Products Directive (TPD) specifies a minimum pack size of 20 cigarettes, having come into force on the 19th May 2014 and subsequently applicable in all EU countries since the 20th May 2016.

(Pages 4-5, lines 91-96)

Page 9: Please elaborate how price differences act to undermine tobacco control efforts, beyond the poor health outcomes associated with tobacco use.

The sentence now reads (new text in bold):

Accurate assessment is important as large price differences undermine tobacco control efforts by negatively influencing quit attempts and smoking initiation, and have been associated with poor health outcomes. It also allows for the identification of tax shifting, including over-shifting (increasing product prices above and beyond the tax). Over-shifting can lead to increases in price gaps as well as industry profits which in turn may be seen as a threat to tobacco control efforts as such resources can serve to maintain the industry’s power to fight public health tobacco control measures. 

(Page 10, lines 267-273)

Reviewer #2: This paper confirms that standardizing pack sizes in the EU was an effective strategy to block promotion through smaller pack sizes. However the authors have tried to pack too much information into the accompanying graphs, making them unreadable.

The authors may wish to make them readable by reducing the amount of information displayed. It is suggested that date for just three years at the beginning middle and end of the study period be displayed. Rather than displaying data for all countries, the point could be made by showing data for one country from each sub-region

Many thanks for this helpful feedback. We believe that the second reviewer’s comments have been addressed through the changes made to the analyses, the inclusion of simplified graphs based on grouped data, and the edits made to the country-level graphs which are now included as Supplementary Material, as described in detail above in response to reviewer one’s comments about the analyses and presentation of the data.

Specifically, we have conducted additional analyses using grouped data from all countries included in the study to display the high-level trends in all price quartiles as well as the cheap and expensive quartiles. These results are presented graphically as the percentage of each pack size category per year (mean across all countries), with the aim of making the graphs more readable by reducing the amount of information displayed, as recommended by the second reviewer also.

However, instead of displaying data for only one country or sub-region, we present the grouped data of all EU countries and provide the country-level graphs in the Supplementary Material in a more interpretable form. By providing both analyses in this way we aim to give the reader a more comprehensive picture of the overall trends at both the EU and country level over the study period.

---

## [Decision Letter · Decision Letter 1]

29 Jul 2020

Size matters: an analysis of cigarette pack sizes across 23 European Union countries using Euromonitor data, 2006 to 2017

PONE-D-19-33505R1

Dear Dr. van Schalkwyk,

We’re pleased to inform you that your manuscript has been judged scientifically suitable for publication and will be formally accepted for publication once it meets all outstanding technical requirements.

Kind regards,

Stanton A. Glantz

Academic Editor

PLOS ONE

Additional Editor Comments (optional):

Reviewers' comments:

Reviewer's Responses to Questions

**Comments to the Author**

1. If the authors have adequately addressed your comments raised in a previous round of review and you feel that this manuscript is now acceptable for publication, you may indicate that here to bypass the “Comments to the Author” section, enter your conflict of interest statement in the “Confidential to Editor” section, and submit your "Accept" recommendation.

Reviewer #1: All comments have been addressed

Reviewer #2: All comments have been addressed

2. Is the manuscript technically sound, and do the data support the conclusions?

Reviewer #1: Yes

Reviewer #2: Yes

3. Has the statistical analysis been performed appropriately and rigorously? 

Reviewer #1: Yes

Reviewer #2: Yes

4. Have the authors made all data underlying the findings in their manuscript fully available?

Reviewer #1: Yes

Reviewer #2: Yes

5. Is the manuscript presented in an intelligible fashion and written in standard English?

Reviewer #1: Yes

Reviewer #2: Yes

6. Review Comments to the Author

Reviewer #1: The authors have vastly improved an already strong paper. The results are much easier to comprehend, the new analysis addresses the aims directly, and their edits throughout the text greatly improve the focus and clarity of their research.

Well done. This is an important study.

Reviewer #2: My main concern in the original review was readability of the graphic material. While the authors have improved readability differently than the way I suggested, the important thing is that readability issues have been satisfactorily addressed.

7. PLOS authors have the option to publish the peer review history of their article (what does this mean?). If published, this will include your full peer review and any attached files.

Reviewer #1: No

Reviewer #2: No

---

## [Editor Report · Acceptance letter]

4 Aug 2020

PONE-D-19-33505R1 

Size matters: an analysis of cigarette pack sizes across 23 European Union countries using Euromonitor data, 2006 to 2017 

Dear Dr. van Schalkwyk:

I'm pleased to inform you that your manuscript has been deemed suitable for publication in PLOS ONE. Congratulations! Your manuscript is now with our production department. 

Kind regards, 

on behalf of

Professor Stanton A. Glantz 

Academic Editor

PLOS ONE